# Percutaneous Ultrasound Guided Celiac Plexus Approach: Results in a Pig Cadaveric Model

**DOI:** 10.3390/ani14233482

**Published:** 2024-12-02

**Authors:** Francesco Aprea, Yolanda Millán, Anna Tomás, Rocío Navarrete Calvo, María del Mar Granados

**Affiliations:** 1Hospital Veterinari Canis Mallorca, 07010 Palma de Mallorca, Spain; 2Department of Comparative Pathology, School of Veterinary Medicine, 14005 Cordoba, Spain; 3Instituto de Investigación Sanitaria de las Islas Baleares (IdISBa), 07120 Palma de Mallorca, Spain; anna.tomas@ssib.es; 4Animal Medicine and Surgery Department, School of Veterinary Medicine, 14005 Cordoba, Spain; nacar6@hotmail.com (R.N.C.); pv2grmam@uco.es (M.d.M.G.)

**Keywords:** regional anaesthesia, interventional analgesia, pigs, celiac plexus, visceral pain, ultra sonography

## Abstract

Visceral pain due to abdominal malignancies or pancreatitis is common in both people and animals. In human medicine, the celiac plexus block or neurolysis are performed in subjects with intractable visceral pain. Ultrasonographic-guided percutaneous techniques for block and/or neurolysis have been described in people. The aim of this study is to describe a percutaneous ultrasound-guided technique to localize the celiac plexus in a pig cadaveric model. The described technique was effective in localizing the celiac plexus in all subjects. Following anatomical and clinical studies, its role in veterinary analgesia should be assessed.

## 1. Introduction

The celiac plexus (CP) is a network of nervous ganglia (celiac, mesenteric and aorticorenal) and interconnecting fibers, and it is the largest visceral plexus. The CP is located in the anterolateral aspect of the abdominal aorta where celiac and mesenteric arteries originate at the thoracolumbar level [1]. The CP receives afferent nociceptive transmissions from the abdominal organs along the autonomic pathway [1].

Since the first report in 1914 [2], the block (CPB) and neurolysis (CPN) of the CP have been employed in human interventional analgesia to control visceral pain in subjects suffering of abdominal malignancies and chronic pancreatitis not responding to conventional analgesic treatment [1,3].

Briefly, CPB and CPN are performed by inserting a needle in proximity to the CP, mostly under fluoroscopy, computed tomography (CT) and ultrasonography (US) guidance. None of these imaging-based techniques have been proven to be superior to another [1,3]. The CPB is primarily used in chronic pancreatitis patients to transiently interfere with the transmission of pain, while the CPN is used primarily in pancreatic cancer for permanent pain relief obtained with ganglion ablation [4].

Abdominal pain is common in veterinary subjects [5,6]. Pancreatitis and pancreatic tumors cause severe visceral pain in small animals, and their treatment is challenging and not based on clinical evidence [5,6,7].

In veterinary medicine, approaches to the CP have been described in standing equines for the management of ileus, and in dogs and swine cadavers for potential clinical applications in clinical pain management [8,9,10]; translational experimental studies describe an endoscopic US-guided (E-US) and a laparoscopic approach to CP in pigs [11,12,13]. Percutaneous US-guided CPN has been reported in humans to treat intractable abdominal pain, but not in veterinary medicine [14,15,16,17,18]. In the last decade, US has been extensively employed in veterinary analgesia for anatomical descriptive studies and clinical trials to control intraoperative nociception and postoperative pain [19,20,21,22,23,24]. The quadratus lumborum block (QLB), a US-guided fascial block, has been described in veterinary species to control surgical visceral nociception [19,20,21,22,23,24]. This block is very promising for perioperative analgesia, but its mechanism of action is not fully understood and currently it is not employed for medical pain management [25].

Interventional analgesic techniques such as CPB and/or CPN (following descriptive studies) could be employed in clinical settings in animals suffering severe visceral pain.

The aim of the study is to describe a percutaneous US-guided approach to the CP in a porcine cadaveric model. To confirm the correct localization of the CP with this technique, stained tissue samples were submitted to histological and immunohistochemical analyses.

## 2. Materials and Methods

This cadaveric study obtained institutional ethical approval (CEEA/005/2020).

### 2.1. Animals

The fourteen fresh swine cadavers included were 3 months old, cross-bred (Duroc cross Large White and Landrace) females, with body masses between 25 and 30 kg. This sample size was based on previously published research [10].

Pigs were enrolled in the study from January 2021 to June 2021. All cadavers were obtained from a not-related, not-survival study and none were euthanized for this project. Cadavers were used immediately following humane euthanasia during general anesthesia with 5 mL pentobarbital 20% applied intravenously (Dolethal, Vetoquinol, Madrid, Spain).

### 2.2. Preliminary Anatomical Description

The location of the CP in this swine strain was established in a previously published cadaveric study [10]. The CP is located retroperitoneally at the level of the last thoracic vertebra (T15) ventrally to the aorta in proximity of the celiac and mesenteric arteries. The correct location of the CP was confirmed by histological evaluation of the sampled tissue [10].

### 2.3. Ultrasound Approach to CP

Cadavers were positioned in right lateral recumbency. A convex transducer (C11, 3s, Mindray M9 GI, Shenzhen, Guangdong, China) was placed on the craniodorsal aspect of the abdomen at the subcostal level just below the last rib (Figure 1). The transducer was gently slid and tilted to identify, in a transverse (short axis) view, the vertebral body of the last thoracic vertebra (T15). Under US guidance, an 18 G (120 mm) Tuohy needle was inserted percutaneously ventrally and parallel to the transverse process of T15 through the last intercostal space (Figure 1). The needle was advanced from dorsolateral to ventromedial through the epaxial muscle and approached the T15 vertebral body (Figure 2). Once the needle tip was in proximity to the lateral aspect of the T15 vertebral body, it was advanced towards the midline (Figure 3). Once the position of the needle was considered appropriate and no resistance to injection was felt, 2 mL of dye (China Ink, Pelikan Drawing Ink, Hanover, Germany) was applied.

### 2.4. Tissue Collection, Processing and Histological and Immunohytochemistry Studies

Through laparotomy, an anatomical dissection of stained tissue was performed and a gross macroscopic examination of the abdominal organ was carried out.

Dyed tissues were handled as previously described [10].

Samples (2 cm, Figure 4) were placed in histological cassettes and fixed in 4% neutral buffered formalin for 24–72 h. Then, 0.5 cm samples were sectioned and embedded in paraffin wax for histological examination. A total of 3 blocks were obtained for each pig. Histochemical study was carried out in 2 serial tissue sections of 4–5 µm from each paraffin block and stained with hematoxilin–eosine (HE).

The CP was identified in the HE-stained tissue sections when clusters of neuron cell bodies encapsulated by ganglionic gliocytes and/or a network of different caliber nerve fibers were observed in at least one out of three histological paraffin blocks per animal.

To confirm the presence of nervous tissue in specific sections already stained with HE, an immunohistochemical study was performed as previously published [10]. Intermediate filament proteins such as neurofilament (NF) and glial fibrillar acid–protein (GFAP) were used with the immunohistochemical methods of complex avidin–biotin–peroxidase.

## 3. Results

### 3.1. Gross Anatomical Description and Ultrasonographic Technique

All fourteen fresh cadavers enrolled completed the study. In all pigs, the same dorsolateral percutaneous US-guided approach was employed. The procedure did not present special challenges in the described settings. Pressure had to be applied to push the abdominal wall with the transducer to reach a sufficiently good sonographic view to allow visualization of the vertebral body (Figure 1). The needle length permitted us to reach the target point (T15 vertebral body).

At dissection following laparotomy, the dye was visible within the retroperitoneal area in proximity to the aorta and celiac vascular trunk (Figure 4). Other retroperitoneal structures including vessels, glands, muscle and fat were stained. Fibers and/or ganglia were not recognizable macroscopically within the stained tissue. The spreading of the dye was approximately from 2 to 4 cm in length. No apparent macroscopic lesions of abdominal organs or vessels were noted at dissection.

### 3.2. Microscopic Studies

In all subjects, the percutaneous US-guided approach was successful in reaching the target structure, since nervous tissue belonging to the CP was observed in 14 out of 14 samples. Specifically in 11 animals, both ganglia and nerves of different sizes were identified—in 2 animals only ganglia, and in 1 cadaver only a network of nerve fibers was found.

Sections of vascular endothelium, striate muscle (maybe from diaphragm or abdominal walls) and glandular tissue (lymphnodes) were observed in some samples together with nervous fibers and ganglia.

The sizes of the ganglia observed ranged from 0.176 mm to 18 mm (Figure 5).

Following HE studies, if only ganglia or nerves were observed, the immunohistochemical technique was employed to confirm the presence of CP. A brown cytoplasmic staining was evident in neuron and nerve fibers with NF and GAFP, respectively.

## 4. Discussion

This cadaveric study describes a percutaneous US-guided approach to the CP in 14 fresh swine cadavers. In all stained samples (14/14) submitted to histological examination, neural tissue belonging to the CP was visible. This finding indicates the efficacy of the technique. This dorsolateral approach to CP has not been previously described in humans or in animals.

The first similar published study [14] described a transcutaneous US-guided approach for CPN in 9 human patients affected by pancreatic carcinoma or chronic pancreatitis. In this report, an anterior (ventral) approach to the CP was used; a 22 G, 15 cm needle was inserted below the xyphoid process. The correct positioning of the needle was confirmed by US using as a landmark the celiac trunk and looking in real time at the spread of the injectate in the preaortic region. Excellent pain relief was achieved in seven of nine subjects, and in five patients analgesia lasted for 6 months. Even though with the anterior approach the needle might pass through the stomach, intestine and pancreas, no complications were observed. Similar techniques were described in other clinical studies in humans affected by inoperable abdominal cancer [14,15,16,17]. In these reports, CPN improved analgesia and reduced systemic analgesics consumption in most of the enrolled subjects. In humans, in a randomized clinical trial, bilateral percutaneous needle insertion was not shown to be superior (in terms of efficacy) to the unilateral anterior approach for US-guided CPN [17]. Visceral analgesia, as described in human patients, could possibly be obtained in veterinary subjects suffering from refractory abdominal pain.

### 4.1. Comparison with Quadratus Lumborum Block (QLB)

In our report, we use a dorsolateral approach with the single-injection technique, with the animal in right lateral recumbency. This technique is a modification of a similar approach described to perform QLB [21,23]. The QLB is a fascial block used in people and in animals to decrease perioperative somatic and visceral pain by blocking spinal nerves and the sympathetic trunk (T13-L3). The use of QLB might be beneficial in perioperative settings, but not for chronic pain management or neurolysis. Its mechanism of action has not been fully elucidated, and due to the nature of the technique (fascial block), differently from CPB, no reports about its use for chronic pain management are available in humans or other species [25].

Different approaches have been described to block QLB in veterinary settings [19,20,21,22,23,24]. When comparing the ventral and lateral approaches with the dorso-lateral (D-QLB) approach, the spread of the dye in cadavers was found to be similar between techniques [19,20,21]. Authors using D-QLB hypothesized that the chances of puncturing abdominal viscera were lower compared to ventral and lateral approaches [21]. The needle was advanced from the dorsolateral to ventromedial through the epaxial muscle, as we did in our study [21].

The CP is located in the retroperitoneal area deeper than QL and psoas muscles; thus, the needle is advanced further and with deeper inclination, aiming at the last thoracic vertebral body. For the QLB, the needle is inserted at the lumbar level, aiming at a more superficial target using a linear ultrasound transducer [21]. When a linear transducer is employed, the anatomy of the most superficial structures is better preserved, but deeper structures are not visualised. To visualize the needle and the vertebral body, we used a convex transducer using an in-plane technique for needle insertion.

For anatomical and technical reasons, the QLB might involve a lower risk of puncturing the diaphragm, organs and major vessels compared to CPB or CPN. Nevertheless, retroperitoneal hematoma has been reported after D-QLB in two dogs [26].

In humans, the rare but severe complications following CPN are weakness, lower chest pain, postural hypotension, dysuria, retroperitoneal abscess, hematuria, peritonitis, and pulmonary embolism [1,27].

### 4.2. Comparison with Other Techniques for CPB and/or CPN

In the last few decades, the CPB and CPN have been performed blinded or guided by fluoroscopy, computed tomography (CT) and ultrasonography (US). In terms of analgesic efficacy, none of these imaging-based techniques have been proven to be superior to any other [1,3].

Posterior (dorsal) approaches to the CP are commonly performed under fluoroscopy and CT guidance to prevent viscera perforation, which is more likely to occur with the anterior (ventral) approach [27]. The technique we have described is similar to a posterior (dorsal) approach, where the needle passes through the epaxial muscles instead of going through the abdominal wall. A dorsal approach under fluoroscopic guidance has been successful in approaching the CP in pig cadavers [10]. The lateral or ventral (supine) position might not be tolerated in bedside interventional procedures in critically ill humans, so anterior approaches are preferred in selected cases [1,27]. With US, differently from other imaging-based techniques, there is no need to use contrast medium (reducing chances of unwanted effects), and it is possible to see in real time the position of the needle and its relationship with target structures and vessels, with no risk for the operator of exposure to radiation. Cost and time are possibly reduced when US is used instead of fluoroscopy or CT.

The use of color flow Doppler in live subjects helps in identifying the aorta and celiac trunk, facilitating the procedure and preventing vascular puncture. Gastroenterologists most commonly perform endoscopic US (E-US) for CPN, using arteries as landmarks for injection [12,13,27].

Two experimental studies carried out with E-US in swine (as translational model) report successful neurolysis in porcine models. This approach to CP is promising, but for technical reasons is not performed by pain practitioners [27].

Recently, a fluoroscopic-guided percutaneous approach used in human interventional analgesia has been described in swine; in 10 out of 12 cadavers, part of the CP was successfully stained [10]. In this report, using US guidance, we successfully approached the CP in all 14 cadavers enrolled. We considered both approaches technically feasible with no challenges and with potential clinical application. However, the data show that the US-guided CP approach seems more successful.

### 4.3. Possible Complications

We did not observe macroscopic lesions in organs or large vessel walls, but we cannot exclude that organ impairment or hemorrhages could occur in live subjects. Due to the nature of the study performed in cadavers, we cannot rule out the occurrence of complications following the technique.

We used an 18 G Touhy needle. The blunt (Huber point) bevel has been studied to provide continuous epidural anesthesia, preventing meningeal puncture [28]. This blunted needle may decrease the chances of accidental vascular puncture, but if this occurs, tissue laceration (and hemorrhages) might be of a greater magnitude than with sharp-tipped needles. A 22 G needle in case of organ or vessel puncture would cause less tissue trauma and injury. A hyperechoic 22 G fine needle would be more appropriate for use in percutaneous techniques in a live subject. A transaortic approach to CP has been described in humans using sharp fine needles [29]. However, the use of an 18 G Touhy needle possibly permits the better appreciation of the loss of resistance when passing through the retroperitoneal paraaortic area. Taking into account the proximity to major vessels, as for other regional techniques, the absence of coagulative disorders should be ruled out prior to the procedure.

### 4.4. Anatomical and Histological Considerations

The CP is composed of one to five ganglia (right and left celiac ganglia, unpaired cranial mesenteric ganglia, and paired aorticorenal ganglia); all of them are connected to each other by a network of nerve fibers [30,31,32,33]. The CP is in the proximity of the diaphragmatic crura, abdominal viscera, adrenal gland, lymph nodes, abdominal wall muscles and large vessels, and it is surrounded by connective and adipose tissues [33]. Proximal structures such as lymphnodes and glands can have similar macroscopic appearances. In clinical settings, pancreatic malignancies can modify the local anatomy, and further affect the accuracy of location [34].

In our study, we could not macroscopically differentiate the CP from surrounding sampled dyed structures (Figure 4b). The stained tissue included adipose cells and occasionally lymphnodes, muscles, adrenal glands or blood vessels. Possibly, the poor dissection of the sampled tissue during laparotomy contributed to these findings. Histochemical studies are paramount in such cases to confirm the presence of nervous tissues [33,35].

Microscopically (with HE), the peripheral nerve is identified in longitudinal or transversal sections as fascicle of nerve fibers (axons) with lightly stained oval nuclei of Schwann cells surrounded by a perineurium; ganglia are formed by varying degrees of densely packed ganglion cells separated by nerve fibers and blood vessels that are commonly large in diameter [33,34,35]. In our study, we could identify both ganglia and nerve fibers of different sizes and a network of interconnected fibers compatible with connection between ganglia (Figure 5). The presence of other tissues (endothelium, muscle, glands) was not surprising considering their anatomical proximity to the plexus [33].

Macroscopically, the ganglia of the plexus are described as V- or C-shaped, triangular, elongated, or rounded [33]. Gerke et al. [36] reported that the size of the plexus ranges from 0.5 to 4.5 cm in humans, which is in accordance with collected samples (0.8–1.8 cm).

Microscopically, we could observe different shapes, and a ganglion composed of several collections of ganglionic cells surrounded by connective tissue. Immunohistochemical methods, utilizing specific antibodies, are employed to distinguish intermediate filaments (IFs) in tissues of unknown origin. Neurofilaments (NFs) and glial fibrillary acidic protein (GFAP) belong to IFs that are structural components of all mammalian cells. The GFAP is expressed mainly in astrocytes, ependymal cells, and Müller cells of the retina, and variable immunoreactivity has been detected in Schwann cells [37,38]. The NFs are always present in proteins of neuronal cells [38]. The use of IFs is common in neoplastic diseases to identify the origin of tumors and to identify nervous tissue when it cannot be identified using routine staining. Other antibodies such as tyrosine hydroxylase are also employed to identify the sympathetic component in the ganglion [33,34,35].

### 4.5. Possible Application to Small Animal

The structure and ultrastructure of the CP have been described in dogs [39], and a percutaneous approach to canine CP has been described [8].

Anatomical differences, such as regarding the shape of the thorax and the number of vertebrae, should be taken into consideration when performing this technique in other species. The typical swine anatomy with an elongated rib cage possibly makes this technique easier to apply in this species compared to in others.

This being a cadaveric study, we did not assess the clinical efficacy of the procedure, but small animals affected by chronic pancreatitis and abdominal malignancies might benefit for CPB and/or CPN for therapeutic or palliative purposes.

We are not able to suggest the optimum volume of local anesthetic or neurolytic agent to be administered to achieve comprehensive visceral analgesia, but large volumes (about 20–30 mL) are mostly employed in humans. In dogs, when a higher volume of local anesthetic has been used for QLB block, it did not result in superior analgesia [40].

We cannot exclude that approaching the first lumbar vertebra instead of the last thoracic one, as described in dogs [8], could lead to similar or superior spreading in the pig.

The feasibility, safety, and efficacy of different approaches should be assessed.

### 4.6. Study Limitations

The absence of ventilation prevented the movement of the diaphragmatic cupula and viscera, which could possibly interfere with this technique in a live subject. Mechanical ventilation of the cadaver could have been used to simulate physiological breathing.

Differently from humans, general anesthesia is likely to be needed to carry out interventional techniques in animals; therefore, the control of ventilation can be performed by the anesthetist, and if considered safe, a brief period of apnea could facilitate the execution of the block. Due to the proximity of the diaphragmatic cupola, the puncturing of the crura is possible, and even though a fine needle would be unlikely to cause major problems, pneumothorax could occur.

General anesthesia will facilitate the procedure, enhancing muscle relaxation; in an awake animal with pain, the pressure applied by the transducer to the cranial abdomen to achieve a good sonographic window might be not tolerated.

A thorough dissection of the entire celiac plexus (including each ganglion and fiber) was not performed. We can only state that part of the plexus (ganglia and fibers) was contacted by the dye; we cannot quantify the degree of blocking of CP that would have been achieved with this amount of injectate. The distributions of the local anesthetic or neurolytic agent differ from those of injected dyes [41]; therefore, further studies are needed to evaluate their distribution. We decided to employ a small volume of dye to avoid significant spreading to adjacent structures, and to facilitate tissue handling, sampling, and analysis, as previously published [10]. We can only suppose that if the CP was stained by a small volume of injectate, it is likely that a larger volume would spread further within the paraaortic retroperitoneal region and stain the entire plexus.

Even though a recently published study showed that food dye was superior to methylene blue and tissue marker for nerve staining [42], we used China ink because this dye was successfully used to stain the CP in previously published studies in dog and swine cadavers [8,10].

## 5. Conclusions

The percutaneous dorsolateral US-guided technique was effective in approaching the CP in swine. The histological evaluation showed a good sensitivity. Anatomical and clinical studies are warranted to establish the role of US-guided CPB and CPN in veterinary analgesia for severe non-surgical visceral pain control.

## Figures and Tables

**Figure 1 animals-14-03482-f001:**
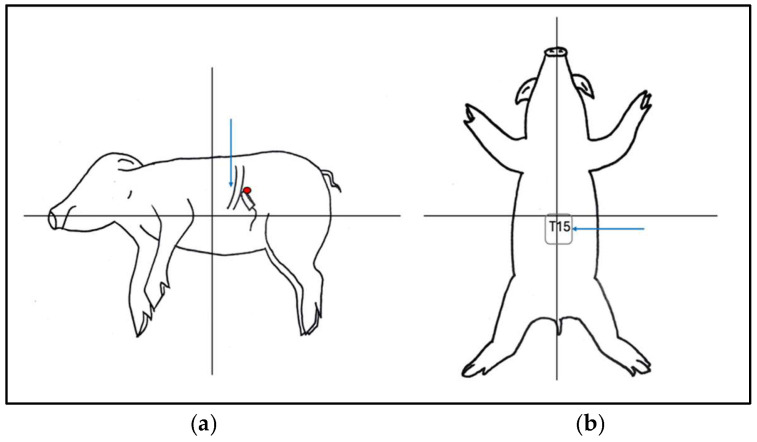
Animals were positioned in right lateral recumbency. The transducer was placed subcostal at the left cranial abdomen underneath the rib cage in a transverse view to identify the last thoracic vertebral body; the red dot indicates the position of the probe and the blue arrow the point of needle insertion (**a**). The blue arrow represents the point of needle insertion at the level of the last intercostal space to approach the vertebral body of T15 (ventral view) (**b**).

**Figure 2 animals-14-03482-f002:**
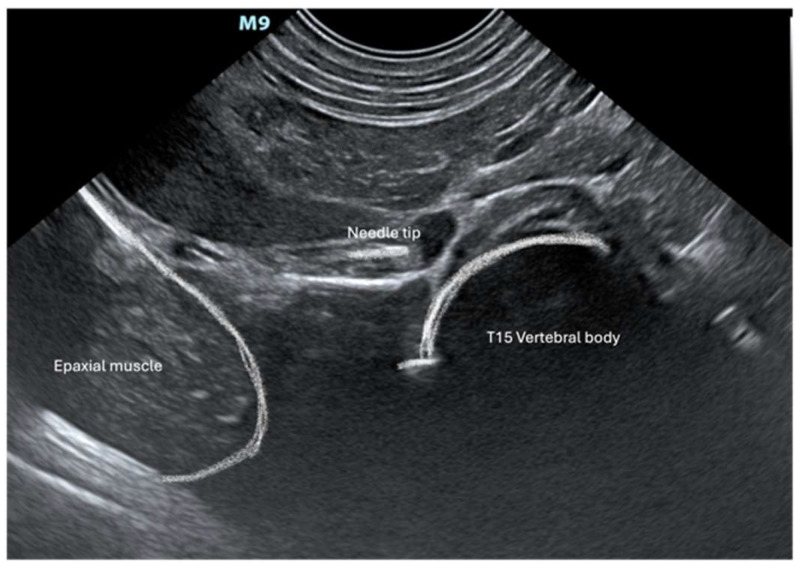
Ultrasonographic image of the left cranial abdomen of the swine cadavers. The needle is inserted through the epaxial muscle, aiming at the vertebral body of the last thoracic vertebral (T15).

**Figure 3 animals-14-03482-f003:**
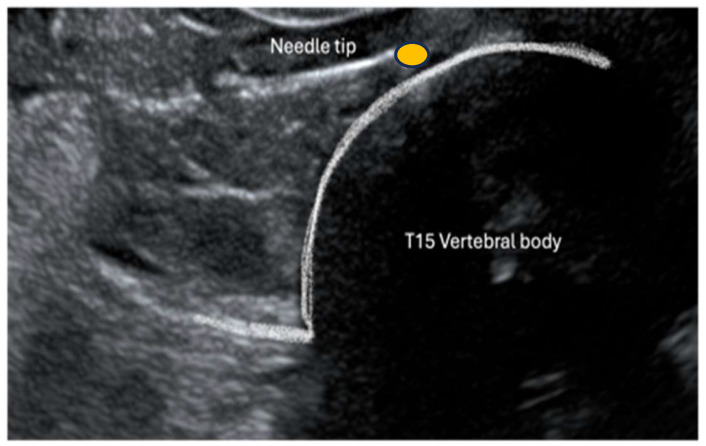
Needle positing prior to injection of dye. The yellow dot represents the CP.

**Figure 4 animals-14-03482-f004:**
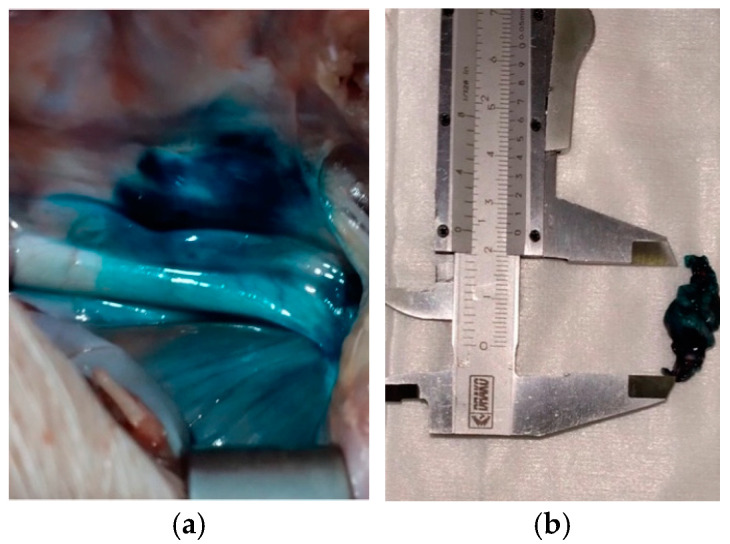
Retroperitoneal stained tissue visible following exploratory laparotomy (**a**) and 2 cm samples dissected for histological preparation (**b**).

**Figure 5 animals-14-03482-f005:**
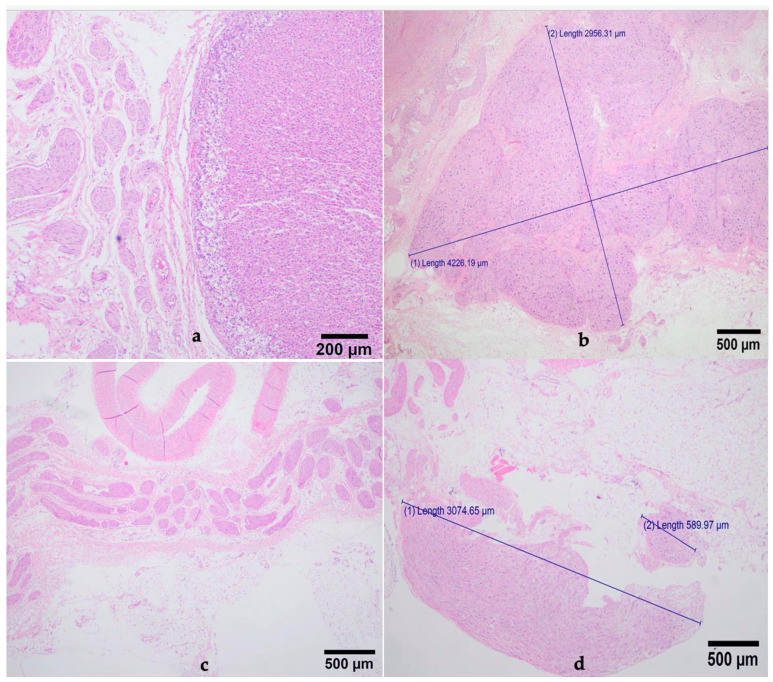
Section of a network of nerve fibers and adjacent glandular tissue (**a**); measures of a ganglia and a nerve fiber (**b**); section of a network of nerve fibers and an adjacent blood vessel (**c**); measures of different sizes ganglia (**d**) (HE).

## Data Availability

Further data are available on request from the corresponding authors.

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
