# Peer review of "Percutaneous Ultrasound Guided Celiac Plexus Approach: Results in a Pig Cadaveric Model"

_animals, 2024, doi:10.3390/ani14233482_

Round 1

Reviewer 1 Report

Comments and Suggestions for Authors

This is a very interesting study and a good continuation of a previous publication by the same authors referenced in this work. I think it would be very nice to expand this research with clinical studies and to evaluate its analgesic response. However, until then, the authors need to be cautious when discussing the clinical application of this CP block, since it was not the aim of this paper. The authors should focus on the anatomy, the technique applied, macroscopical evaluation of the injection and the histological aspects, and relate the results obtained with the results of other studies in the swine or other species including humans. More specific comments on the pdf attached/ below.

Author Response

Reviewer 1

We really appreciate you consider it an interesting study. Thank you so much for your advice. We will focus on the underlined suggestions. Changes are marked in red.

ABSTRACT

Line 22: the sentence has been modified

Line 33: the article has been added

Introduction

Line 69: this sentence has been removed

Line 81-82: Thank you to point this out. A sentence and a reference have been added.

Material and Methods

Thank you to point out, as you correctly state we decided to choose this sample sized based on our previous experience. We added this information within the text.

Line 94: the text has been modified

Results

Line 145: thank you to point out this, the text has been modified and information added as suggested

Discussion

Line 212: thanks, the sentence has been rephrased

Line 213: the text has been modified as suggested

Line 222-223: the text has been modified as indicated

Line 228: changed

Line 232: thanks to point this out, we have changed the order of the sentence

Line  239: the text has been modified

Line 240: thank you to point this out, the text has been amended according

Complications reported are in general, in terms of complications there is not an approach considered safer than another but in terms of anatomical structures to be encountered a dorsal approach seems safer because there are no organs in the way.

Line 243: the paragraph has been moved as suggested

Line 249: thanks to point this out, it is bedside procedure

Line 256: it has been clarified in the text

Line 262: references have been added

Line 268: it is correct to say “fluoroscopic guided” even though expression such as fluoroscopy guided and fluoroscopically guided are employed too.

Line 271: thanks to point this out, the text has been modified

Line 274: this sentence has been removed

Line 278: thanks, it was added in the M&M

Line 279: the text has been modified as suggested

Line 281: this sentence was moved above in the text

Line 334-344: this text has been deleted as suggested.

Line 359: it has been added

Line 373: the text has been amended as suggested

Line 380: the sentence has been modified

Line 387: we linked the two sentences

Line 392: this phrase has been removed

Figure 3: the dot has been added, thank you,

Thank you so much to help us to improve the quality of our manuscript

Reviewer 2 Report

Comments and Suggestions for Authors

The manuscript is well structured, clear and well written. I have the following corrections/suggestions:

Line 16: Delete “and” … for block or neurolysis.

Line 65: “…transmission of pain controlled…” Delete controlled or rephrase this sentence.

Line 69: I suggest rewrite this sentence starting a new phrase  as “Furthermore, validated tools….”.

Line 76. Replace “setting” for medicine.

Line 77. “clinical trials”.

Line 82. “in clinical setting in veterinary subjects” can be replaced by just one word: “animals”.

Line 93. The letter “p” of Pigs should be written in capital.

Line 95. The route of administration should be stated. Were pigs under general anaesthesia?

Line 131. I do not understand the meaning of  “at least from 1 to 3 block per animals”

Line137. The word development could be deleted

Line 140. What do you mean by pressure? How was this pressure produced?

Line 165. Animals were positioned…

Line 174. Replace “window” for image and “cadaver” for cadavers.

Lines 226-231. This paragraph may not be relevant for your study. Perhaps it could be deleted.  

Line 243. “have been reported” can be deleted.

Line 261/279. in alive subjects.

Line 318. “or?? C-shaped” Please revise this words.

Line 320. You should mention the species were a range of 0.5 to 4.5 cm was described. I assume it was not in pigs.

Line 325. You should define “IFs”

Line 395. I do not understand in the context of your work the meaning of “good sensitivity”.  

Author Response

Thank you so much for your comments, we are glad you considered the manuscript clear and well written.

Changes in the text are marked in blue.

Line 16: the text has been modified

Line 65: the text has been amended as suggested

Line 69: this part has been deleted accordingly to the other reviewer comment

Line 76: the text has been modified

Line 82: the text has been changed as suggested

Line 95: thanks to point this out, this info has been added

Line 131: it refers to paraffin blocks, the text has been modified to make it clear to the reader

Line 137: the title has been modified

Line 140 : Thanks to point this out, the sentence has been changed to make the text clear for the reader

Line 226-231: we used a similar approach to the one described for the D-QLB for this reason we would like is possible to keep this sentence

Line 243 : it has been deleted

Line 261/279: the text has been modified

Line 328: the text has been changed

Line 325: it was defined in line 324 (intermediates filaments)

Line 395: we mean that the histological technique chosen was useful to identify the target structure. If you prefer, we can rephrase this sentence

Thank you so much for your suggestions.

Reviewer 3 Report

Comments and Suggestions for Authors

Dear Autors

thank you for your work. 

You can find  my comments in the attachment

kind regards

Author Response

We are very thankful for your comments. Changes in the text are written in yellow.

Line 126: the text has been modified so it is now clear for the reader

Fig 2 : We use an in plane approach but we employed a micro convex probe not a linear probe so the needle was not visible throughout all is length 

Line 297: The CP is composed of 1 to 5 ganglia; the text has been modified

Line 318: the text has been corrected

Line 320: it is in humans; it has been added in the text

Thank you so much to help us improving the quality of our manuscript.

Round 2

Reviewer 1 Report

Comments and Suggestions for Authors

Well done and thank you for the amendments made

Last minor comments mainly spelling

Line 116: The needle was advanced from dorsolateral to ventromedial through the epaxial muscle and approached T15 vertebral body…

Line 135: per animal

Line 222: patients

Line 223: suffering from

Line 249: ….to CPB or CPN. Nevertheless,  retroperitoneal…..

Author Response

Dear reviewer,

thank you so much for your comments. The suggested changes have been made in the text. We are really thankful for your help.